# An Online Calibration Method for a Galvanometric System Based on Wavelet Kernel ELM

**DOI:** 10.3390/s19061353

**Published:** 2019-03-18

**Authors:** Wugang Zhang, Wei Guo, Chuanwei Zhang, Shuanfeng Zhao

**Affiliations:** School of Mechanical Engineering, Xi’an University of Science and Technology, Xi’an 710054, China; guow@xust.edu.cn (W.G.); zhangcw@xust.edu.cn (C.Z.); zsf@xust.edu.cn (S.Z.)

**Keywords:** two-dimensional galvanometer, online calibration, wavelet kernel extreme learning machine

## Abstract

The online calibration method of a two-dimensional (2D) galvanometer requires both high precision and better real-time performance to meet the needs of moving target position measurement, which presents some challenges for traditional calibration methods. In this paper, a new online calibration method is proposed using the wavelet kernel extreme learning machine (KELM). Firstly, a system structure is created and its experiment setup is established. The online calibration method is then analyzed based on a wavelet KELM algorithm. Finally, the acquisition methods of the training data are set, two groups of testing data sets are presented, and the verification method is described. The calibration effects of the existing methods and wavelet KELM methods are compared in terms of both accuracy and speed. The results show that, for the two testing data sets, the root mean square errors (RMSE) of the Mexican Hat wavelet KELM are reduced by 16.4% and 38.6%, respectively, which are smaller than that of the original ELM, and the standard deviations (Sd) are reduced by 19.2% and 36.6%, respectively, indicating the proposed method has better generalization and noise suppression performance for the nonlinear samples of the 2D galvanometer. Although the online operation time of KELM is longer than ELM, due to the complexity of the wavelet kernel, it still has better real-time performance.

## 1. Introduction

A two-dimensional (2D) galvanometer is an important optical component. Due to its compact structure, small driving load, and fast response speed, it is widely used in material processing [1,2,3,4], laser projection [5,6,7], optical measurement [8,9], biomedical imaging [10,11], automatic vehicle driving [12], optical communication [13], and other aspects.

The photoelectric measurement system with a 2D galvanometer must have fast measuring speed and high measuring precision when tracking and positioning the moving target in real time. As shown in Figure 1, the measurement system is mainly composed of a 2D galvanometer and a laser rangefinder. In the three-dimensional measurement space, the maneuvering target moves along the unknown curve C to the ith position, $P_{i}(i=1,2,\ldots ,N)$, and the laser beam of the range finder is refracted by the mirrors of the galvanometer. When the laser spot is following the moving object, the system detects the rotation angles of mirrors, $\left(\theta_{x},\theta_{z}\right)$, and the distance, D, and then obtains the measured value, $M_{i}(D_{i}^{p},\theta_{xi}^{p},\theta_{zi}^{p})$, of $P_{i}$. According to the geometric relationship of the galvanometer, the position of the moving target can be calculated online. The geometric structure is analyzed with the following relations:(1)$$D_{i}^{p}=P_{a}P_{b}+P_{b}P_{c}+P_{c}P=l+P_{b}P_{c}+P_{c}P_{i},$$
(2)$$OP_{b}=OP_{d}=e,$$
(3)$$P_{b}P_{c}=P_{d}P_{c}.$$

The galvanometer detection value, $M_{i}(D_{i}^{p},\theta_{xi}^{p},\theta_{zi}^{p})$, and the corresponding position coordinates, $P_{i}(x_{i}^{p},y_{i}^{p},z_{i}^{p})$, in the measurement space satisfy the following nonlinear matrix equation: (4)$$\left[\begin{array}{c} x_{i}^{p}\\ y_{i}^{p}\\ z_{i}^{p} \end{array}\right]=f_{1}\left[\begin{array}{c} D_{i}^{p}\\ \theta_{xi}^{p}\\ \theta_{zi}^{p} \end{array}\right]=(D_{i}^{p}-l)\cdot \left[\begin{array}{c} \cos2\theta_{zi}^{p}\cdot \sin2\theta_{xi}^{p}\\ \cos2\theta_{zi}^{p}\cdot \cos2\theta_{xi}^{p}\\ \sin2\theta_{zi}^{p} \end{array}\right]+\left[\begin{array}{c} -e\cdot \sin2\theta_{xi}^{p}\\ -e\cdot \cos2\theta_{xi}^{p}\\ 0 \end{array}\right].$$

The mathematical relationship defined in Equation (4) is called the ideal physical model of the 2D galvanometer. Obviously, there is a nonlinear relationship between the coordinates and the detection value.

Due to the influence of installation error, system control error, laser ranging error, angle measurement error, and random noise, there is usually a large error between the actual position and theoretical position calculated by the ideal physical model. Obviously, in order to measure the moving target position in real time, an online calibration method is needed, which can correct the measurement error, caused by many factors, to improve the online measurement accuracy.

Currently, the calibration methods mainly include model-driven [14,15,16] methods and data-driven methods. The model-driven methods are established on the basis of the ideal physical model method shown in Equation (4). If the working range of the 2D galvanometer is concentrated near the center of the optical axis, higher accuracy can be obtained by using the physical model calibration method [17,18]. If the working range is far away from the center of the optical axis, various errors and random noise must be considered. Alkhazur Manakov et al. [19] presented a complicated model containing up to 26 physical parameters to predict the distortions of the 2D galvanometer. Even so, not all affecting factors are involved. Too many parameters increase the complexity of modeling and lead to the risk of local minimization [20]. The modeling difficulty can be reduced by improving the hardware precision [21], but this will greatly increase the cost.

With the data-driven method, these deficiencies can be well avoided. As shown in Figure 1 and Figure 2, for the arbitrary moving target $G_{i}(x_{i}^{G},y_{i}^{G},z_{i}^{G})$ in the measurement range, its corresponding measurement value is $M_{i}^{G}(D_{i}^{G},\theta_{xi}^{G},\theta_{zi}^{G})$. A fixed calibration plane, $O_{1}x_{1}z_{1}$, can be set in front of the galvanometer, the collinear point with $OG_{i}$ in $O_{1}x_{1}z_{1}$ is $P_{i}(x_{i}^{p},y_{i}^{p},z_{i}^{p})$, and its measurement value is $M_{i}^{P}(D_{i}^{p},\theta_{xi}^{p},\theta_{zi}^{p})$. Due to $\theta_{xi}^{p}=\theta_{xi}^{G}$ and $\theta_{zi}^{p}=\theta_{zi}^{G}$, the following relation can be obtained from Equation (4):(5)$$\{\begin{array}{c} x_{i}^{G}=\frac{D_{i}^{G}-l}{D_{i}^{p}-l}\cdot x_{i}^{p}\\ y_{i}^{G}=\frac{D_{i}^{G}-l}{D_{i}^{p}-l}\cdot y_{i}^{p}\\ z_{i}^{G}=\frac{D_{i}^{G}-l}{D_{i}^{p}-l}\cdot z_{i}^{p} \end{array}.$$

Theoretically, if the measured values of all the position points in the calibration plane are known, the coordinates of each position in the measurement space can be obtained through Equation (5).

The data-driven calibration algorithm selects a limited number of points in the calibration plane to establish the data set $\left\{t_{i}|i=1,2,\ldots ,N\right\}$, obtains the corresponding measurement values of each point through experiments to establish the measurement data set $\left\{x_{i}|i=1,2,\ldots ,N\right\}$, and uses a machine learning algorithm to establish a nonlinear mapping model, f, between the two groups of data sets as the calibration model. For any measured value, $M_{i}$, that does not belong to the $\left\{x_{i}|i=1,2,\ldots ,N\right\}$, the mapping model f is used to obtain the corresponding position $P_{i}$ in the $O_{1}X_{1}Z_{1}$ plane.

Stefan et al. [22] used artificial neural networks (ANN) to analyze the measurement data of the galvanometer and established the calibration model. In order to improve the learning effect of the artificial neural network, ridge regression and regularization coefficients [23] are introduced to avoid over-fitting of the network, and higher calibration accuracy is obtained. Tobias Wissel et al. [24] made a comparative analysis of three data-driven methods as follows: ANN, support vector regression (SVR), and Gaussian processes (GPs), and the result shows that the GPs method could obtain higher precision but slow convergence. The learning speed of ANN and SVR was faster than GPs and the accuracy of the three data-driven methods was better than that of the model-driven method. However, as ANN and SVR methods use an offline training mode to establish a calibration model, those methods often require long training time.

The extreme learning machine (ELM) is a single hidden layer neural network, which has the advantages of fewer training parameters, faster training speed, and strong generalization ability [25,26]. Its learning speed is many times faster than ANN, SVR, and least squares SVR algorithms [27] and it is helpful to the solution of the online calibration problem. However, the ELM algorithm uses linear weighted mapping to train the calibration data sets. For nonlinear samples, the accuracy of ELM is reduced [28]. In addition, the ELM algorithm randomly selects the weights and uses the trial and error method to determine the optimal number of neurons, which will affect the stability and rigor of the calibration algorithm.

For this reason, Huang, et al. [27] proposed the kernel extreme learning machine (KELM), which utilized the kernel function to replace the linear weighted mapping in ELM. Kernel function can implicitly map the nonlinear training samples to the high-dimensional feature space and use the linear function to make nonlinear training samples more easily structured. KELM replaces the inner product in the linear algorithm with a kernel function to obtain an optimal least square solution. As a result of this, it has fast training speed and strong generalization ability. Therefore, in this paper, KELM is proposed to train the calibration data of the galvanometer with nonlinear characteristics and to solve the online calibration problem. For the given calibration data, different kernel functions have different capabilities of mapping, resulting in significant differences in the generalization. Therefore, choosing the kernel function suitable for galvanometer calibration is the key to solving the problem. The original kernel functions mainly include a polynomial kernel, Sigmoid kernel, Gaussian kernel, etc. Normally, the wavelet function has better characteristics of multi-scale subdivision and noise suppression. If the kernel function could be constructed on the basis of the mother wavelet, the algorithm would have a superior performance. Due to the complex nonlinear characteristics of the calibration data, it is difficult to determine the appropriate kernel from prior knowledge. The two wavelet kernel functions of the Morlet wavelet and Mexican Hat wavelet have been proven to be the admissible ELM kernel [29,30,31]. Therefore, cross-validation will be used to verify which kernel function is best for galvanometer calibration. This article uses two groups of testing data sets to verify the training results and the results are compared and analyzed with the physical model method and the original ELM. The kernel functions with minimum error are considered to be more suitable for online calibration. On this basis, the reasons for why the wavelet KELM calibration method performs better than other methods are further analyzed from the aspects of generalization ability, noise suppression, and training time.

In this paper, a data-driven approach for the online calibration of a 2D galvanometer is presented. In Section 2, the principle of galvanometer calibration and the experimental device is described. In Section 3, the wavelet KELM algorithm is introduced, and the process of establishing the online calibration method is analyzed. In Section 4, the experimental scheme and verification method are described. Finally, the conclusion is given in Section 5 and the experimental results of existing calibration methods and wavelet KELM method are compared and analyzed.

## 2. System Construction and Experiment Device

### 2.1. System Calibration Description

The experimental device for galvanometer calibration of the photoelectric measurement system, which is composed of a 2D galvanometer measurement system, a 2D moving platform, and a photoelectric target, is shown in Figure 3. The galvanometer measurement system is composed of the galvanometer units, laser rangefinder, controller, computer, and so on. Each set of galvanometer units is composed of a motor, an encoder and a mirror. The axes of the galvanometer units are perpendicular to each other. The laser beam emitted by the rangefinder is reflected to the surface of the photoelectric target after being refracted twice by the mirrors. A part of the light is reflected by the photoelectric target and is captured by the laser range finder. The position of the laser spot can be adjusted by the rotation of mirrors that are driven by the servo system under the control of the computer.

The system takes the intersection point of the x motor rotation axis and the mirror center as the coordinate origin. When the two encoders are located at the zero position, the *Y*-axis is parallel to the direction of the outgoing laser beam and the vertical direction is the *Z*-axis. Then, the measurement coordinate system is established, which is denoted as O−XYZ. The 2D moving platform has two motions, *X*-axis and *Z*-axis, and is composed of the computer, controller, stepper motors, mechanical components, and other parts. The photoelectric target is composed of a CCD camera, projection plate, and a spot location detection module and it is installed on the table of *X*-axis. When the laser spot is projected on the surface of the photoelectric target, the CCD detects the spot, identifies and calculates the deviation, $\delta (u_{p},v_{p})$, between the spot center and the photoelectric target center, and feeds it back to the galvanometer controller through the wireless data module. The measuring system uses the spot position deviation $\delta (u_{p},v_{p})$ as the feedback signals to control the galvanometer. In this way, the spot can accurately follow the movement of the photoelectric target accurately.

When the photoelectric target moves to the ith (i=1,2,…,N) position, the coordinates of the photoelectric target, $Q_{i}(x_{i}^{q},y_{i}^{q},z_{i}^{q})$, and the deviation, $\delta_{i}(u_{i}^{p},v_{i}^{p})$, can be detected and the corresponding coordinates of the spot center, $P_{i}(x_{i}^{p},y_{i}^{p},z_{i}^{p})$, can be calculated by the following equation:(6)$$\{\begin{array}{l} x_{i}^{p}=x_{i}^{q}+u_{i}^{q}\\ y_{i}^{p}=y_{i}^{q}\\ z_{i}^{p}=z_{i}^{q}+v_{i}^{q} \end{array}.$$

The center coordinate of the spot, $P_{i}(x_{i}^{p},y_{i}^{p},z_{i}^{p})$, is represented as the vector $t_{i}={\left[x_{i}^{p},y_{i}^{p},z_{i}^{p}\right]}^{T}$ and the measurement value corresponding to the point $P_{i}$ is represented as the vector $x_{i}={\left[D_{i}^{p},\theta_{xi}^{p},\theta_{zi}^{p}\right]}^{T}$, where $\left(\theta_{xi}^{p},\theta_{zi}^{p}\right)$ are the encoder values of mirror rotation and $D_{i}^{p}$ is the laser ranging value.

Through experiments, the galvanometer calibration data set $\left\{(x_{i},t_{i})|i=1,2,\ldots ,N\right\}$ with *N* samples is established. The essence of the data-driven calibration method is to seek the nonlinear mapping $f:x_{i}\to t_{i}$ from input $x_{i}$ to output $t_{i}$. The purpose of galvanometer calibration is to use the KELM algorithm to establish the mapping model $f:x_{i}\to t_{i}$ as the calibration model for the training data set.

For any measurement value $x_{i}$, the calibration value of the spot position $t_{i}^{Model}$ can be calculated by the calibration model, so as to satisfy the following inequalities: (7)$$\sqrt{\frac{1}{N}{\sum {}_{i}^{N}\left\|t_{i}^{Model}-t_{i}\right\|}^{2}}<\varepsilon$$
where ε is calibration accuracy preset by the system.

### 2.2. Experiment Device

Figure 4 illustrates the experimental device of galvanometer calibration for the photoelectric measurement system. It uses the SKD-100D laser rangefinder (Sankoe, Xi’an, China), with a maximum measuring distance of 100 meters and a measurement accuracy of ±1 mm. The two galvanometer units of the 2D galvanometer each contain a 21-bit encoder and an AC servo motor. Both galvanometers are controlled by a MPC08D board (Leetro, Chengdu, China). The measuring range of the galvanometer is $\pm 16^{\circ }$ of X-scanner and $\pm 14^{\circ }$ of Z-scanner. The photoelectric target uses an MV-EM200M industrial camera (Microvision, Beijing, China) with a 1/1.8”CCD. The actual length of the projection plate corresponding to each pixel is 0.06 mm. The motion controller uses the open-loop controlling mode to control the two stepper motors dragging the moving platform. The repetitive positioning accuracy of the moving platform is less than 0.2 mm. The effective stroke of the moving platform is 1.5 m for the *X*-axis and 1.2 m for the *Z*-axis.

## 3. Mathematical Model and Calibration Method

### 3.1. Kernel Extreme Learning Machine (KELM)

For the galvanometer calibration data set $\left\{(x_{i},t_{i})|i=1,2,\ldots ,N\right\}$ with *N* samples, the ELM algorithm can be used to establish the mapping model $f:x_{i}\to t_{i}$ as the calibration model. The standard single hidden layer neural network with L hidden nodes can be used to establish the mapping model $f:x_{i}\to t_{i}$, which can be expressed as follows:(8)$$f(x)=h(x)\beta =\sum_{i=1}^{L}\beta_{i}g_{i}(w_{i}\cdot x_{j}+b_{i})=t_{j},(j=1,\ldots ,N)$$
where g(x) is the activation function, $w_{i}={\left[w_{i1},w_{i2},w_{i3}\right]}^{T}$ is the weight vector to connect the ith hidden node and the input node, $\beta_{i}={\left[\beta_{i1},\beta_{i2},\beta_{i3}\right]}^{T}$ is the weight vector to connect the ith hidden node and the output node, and $b_{i}$ is the threshold of the ith hidden node. The $w_{i}\cdot x_{j}$ denotes the inner product of $w_{i}$ and $x_{j}$. The above equation can be written as follows:(9)Hβ=T
where
(10)$$H(w_{1},\ldots ,w_{L},b_{1},\ldots ,b_{L},x_{1},\ldots ,x_{N})={\left[\begin{array}{ccc} g(w_{1}\cdot x_{1}+b_{1}) & \cdots & g(w_{L}\cdot x_{1}+b_{L})\\ \vdots & \cdots & \vdots \\ g(w_{1}\cdot x_{N}+b_{1}) & \cdots & g(w_{L}\cdot x_{N}+b_{L}) \end{array}\right]}_{N\times L},$$
(11)$$\beta ={\left[\begin{array}{c} \beta_{1}^{T}\\ \vdots \\ \beta_{L}^{T} \end{array}\right]}_{L\times 3}^{}T={\left[\begin{array}{c} t_{1}^{T}\\ \vdots \\ t_{N}^{T} \end{array}\right]}_{N\times 3}^{}.$$

According to the proof process given by Huang et al. [25,26], the least-squares solution of the general linear system can be represented as:(12)$$\beta =H^{†}T$$
where *H*^†^ is the Moore–Penrose generalized inverse of matrix *H.*

The traditional ELM algorithm is based on the empirical risk minimization criterion and there is a risk of over-fitting in the training process. Therefore, output weights β can be obtained by finding the least square solution of the following problem: (13)$$\mathrm{Minimize}:\;L_{PELM}=\frac{1}{2}{\left\|\beta \right\|}^{2}+C\frac{1}{2}\sum_{i=1}^{N}\xi_{i}^{2}\;\mathrm{Subject}\;\mathrm{to}:\;h(x_{i})\beta =t_{i}-\xi_{i}i=1,2,\ldots ,N.$$
where $h(x_{i})$ is the ith hidden-layer output vector, $\xi_{i}$ is the difference between the ith sample and the output of the hidden layer. Based on the Karush–Kuhn–Tucker theorem, the solution of the above quadratic optimization problem is equivalent to solving the Lagrange function problem as follows:(14)$$L_{DELM}=\frac{1}{2}{\left\|\beta \right\|}^{2}+C\frac{1}{2}\sum_{i=1}^{N}\xi_{i}^{2}-\sum_{i=1}^{N}\alpha_{i}(h(x_{i})\beta -t_{i}+\xi_{i}),$$
the output weight is
(15)$$\hat{\beta }=H^{T}{\left(\frac{1}{C}+HH^{T}\right)}^{-1}T,$$
the output function is
(16)$$f(x)=h(x)H^{T}{\left(\frac{1}{C}+HH^{T}\right)}^{-1}T.$$

The linear weighted hidden output function h(x) is unknown and usually does not satisfy the nonlinear sample mapping. In order to improve the fitting ability of the algorithm to nonlinear samples, the kernel function κ(u,v) can be used to replace $h(x)H^{T}$ and $HH^{T}$ in Equation (16). The output function is as follows:(17)$$f(x)={\left[\begin{array}{c} \kappa (x,x_{1})\\ \vdots \\ \kappa (x,x_{N}) \end{array}\right]}^{T}{\left(\frac{1}{C}+\Omega_{ELM}\right)}^{-1}T$$

Where $\Omega_{ELM}$ is kernel function matrix as follows:(18)$$\Omega_{ELM}=h(x_{i})\cdot h(x_{j})=\kappa (x_{i},x_{j}).$$

According to Equations (16)–(18), if the kernel function κ(u,v) and regularization coefficient C are determined, the corresponding network output can be obtained.

### 3.2. Translation-Invariant Kernel Theorem

The selection of kernel function has a great influence on the accuracy of KELM. With different kernel functions, the accuracy of calibration varies greatly. If a binary function κ(u,v) satisfies Mercer’s theorem, it is an admissible kernel function. In fact, it is difficult to prove that a binary function satisfies Mercer’s theorem. Fortunately, for the translation-invariant kernel function, the following theorem provides a necessary and sufficient condition for it become an admissible kernel function.

Theorem 1 (translation-invariant kernel theorem) [32]

A translation-invariant kernel $\kappa (x_{i},x_{j})=\kappa (x_{i}-x_{j})$ is an admissible kernel, if and only if the following Fourier transform:(19)$$F[K](\omega )={\left(2\pi \right)}^{-D/2}\int_{R^{D}}\exp(-j\omega x)K(x)dx$$
is non-negative.

Traditional kernel functions mainly include a linear kernel, Sigmoid kernel, polynomial kernel, Gaussian kernel, etc. The commonly used translation-invariant kernel functions are Gauss kernel functions and polynomial kernel functions. Since the linear kernel and Sigmoid kernel are not suitable for the fitting of nonlinear samples, this paper focuses on the polynomial kernel function and the Gaussian kernel function. The expression of the two kernel functions can be given as follows:(20)$$\mathrm{Polynomial}\;\mathrm{kernel}:\;\kappa (x_{i},x_{j})={\left(x_{i}^{T}x_{j}+c\right)}^{d},$$
(21)$$\mathrm{Gaussian}\;\mathrm{kernel}:\;\kappa (x_{i},x_{j})=\exp(-\frac{{\left\|x_{i}-x_{j}\right\|}^{2}}{2\sigma^{2}})$$
where *d* is an adjustable polynomial power exponent and σ is Gaussian core width.

### 3.3. Wavelet Kernel Function

The mapping accuracy will be affected by the noise and nonlinear characteristics of the calibration data set of the 2D galvanometer. Therefore, the performance of the kernel function is particularly important. For 2D galvanometer calibration, the kernel function is required to have both good nonlinear mapping capability and certain noise suppression capability.

In general, the wavelet function has better characteristics of multi-scale subdivision and noise suppression and, if the kernel function can be constructed by the wavelet function, it is expected to construct an ideal kernel for KELM to solve the galvanometer calibration problem. It can be proved that the Morlet wavelet kernel function and the Mexican Hat wavelet kernel function satisfy the translation-invariant kernel theorem and are admissible ELM kernel functions [29,30,31]. The proof process will not be repeated in this paper. The two wavelet kernel functions can be given as follows: (22)$$\mathrm{Morlet}\;\mathrm{wavelet}\;\mathrm{kernel}:\;\kappa (x_{i},x_{j})=\cos(a\cdot \frac{\left\|x_{i}-x_{j}\right\|}{\sigma })\exp(-\frac{{\left\|x_{i}-x_{j}\right\|}^{2}}{2\sigma^{2}}),$$
(23)$$\mathrm{Mexcian}\;\mathrm{Hat}\;\mathrm{wavelet}\;\mathrm{kernel}:\;\kappa (x_{i},x_{j})=(1-\frac{{\left\|x_{i}-x_{j}\right\|}^{2}}{\sigma^{2}})\exp(-\frac{{\left\|x_{i}-x_{j}\right\|}^{2}}{2\sigma^{2}})$$
where a is non-negative constant coefficient and σ is non-negative wavelet parameter.

In this paper, the calibration results of the Morlet wavelet kernel and the Mexican Hat wavelet are compared with other methods in Section 5 to further investigate the performance of wavelet kernel functions.

### 3.4. Online Calibration Method

In this section, the algorithm flow of online galvanometer calibration is presented. For the 2D galvanometer device, the main parameters that affect the calibration accuracy are considered to be constant values. That is to say, the nonlinear characteristics of the galvanometer distortion are time-invariant. Therefore, the online calibration program can be divided into an offline training part and online calculation part. The time-consuming model training is scheduled to be completed in the offline stage and, after the training is finished, the parameters are saved. In the online calculation stage, the parameters are read in and the model is built rapidly, then the measured values can be put into the model to quickly calculate the calibrated results.

The calibration process is divided into part A and part B. Part A is the offline model training, and part B is for online calculation. The online calibration-based KELM algorithm is as follows:

Part A: Model training

Step (1): Obtain the training data set $\left\{(x_{i},t_{i})|i=1,2,\ldots ,N\right\}$ of galvanometer calibration and the testing data set $\left\{(x_{i}^{t},t_{i}^{t})|i=1,2,\ldots M\right\}$, initialize the regularization coefficient C and other parameters in the kernel function.

Step (2): Put $x_{i}$ into the Equations (20)–(23), and compute the kernel matrix $\Omega_{ELM}$.

Step (3): Calculate the output weight matrix: $O_{w}={\left(\frac{1}{C}+\Omega_{ELM}\right)}^{-1}\cdot t_{i}$.

Step (4): According to Equation (17), the output f(x) is calculated and compared with $t_{i}$ to obtain the training accuracy.

Step (5): Put $\left\{x_{i}|i=1,2,\ldots ,N\right\}$ and $\left\{x_{i}^{t}|i=1,2,\ldots ,M\right\}$ in the Equations (20)–(23) and compute the kernel matrix $\Omega_{Test}$.

Step (6): The kernel matrix $\Omega_{Test}$ and the output weight $O_{w}$ are substituted into Equation (17) and the output f(x) is calculated and compared with $\left\{t_{i}^{t}|i=1,2,\ldots ,M\right\}$ to obtain the test accuracy.

Step (7): By means of cross-validation, the regularization coefficient *C* and kernel function parameters are modified until the optimal training accuracy is obtained, and then the optimal output weight, regularization coefficient, and kernel function parameters are saved and the network training is completed.

Part B: Online calculation

Step (1): Read the optimal output weight matrix, regularization coefficient, and other kernel function parameters.

Step (2): Read a set of measurements value $x_{i}={\left[D_{i},\theta_{xi},\theta_{zi}\right]}^{T}$ online, put them in the Equations (20)–(23), and compute the kernel matrix $\Omega_{Test}$.

Step (3): The kernel function matrix $\Omega_{Test}$ and output weight $O_{w}$ are substituted into Equation (17) to calculate the output f(x) and the calibration value of the spot center position corresponding to the measured value is obtained.

Step (4): Return to Step (2) to calculate the next laser spot coordinates.

## 4. Experiments and Methodology

The training data set and two groups of testing data sets are collected by using the experimental device of Figure 4. The method of calibration data generation is shown in Figure 5. The 2D moving platform is set up at positions I and II, respectively. At position I (OO_1_ = 1.2 m), the training data set and the first testing data set are generated. At position II (OO_2_ = 2.4 m), the second testing data set is generated. The data generation method is described in detail in the following section.

### 4.1. Training Data Set Generation and Cross Validation

As shown in Figure 5, the 2D moving platform is set at position I and the platform is parallel to the coordinate plane OXZ. The plane $O_{1}X_{1}Z_{1}$ is called the calibration plane. In $O_{1}X_{1}Z_{1}$, the 2D moving platform drives the photoelectric target along X or Z directions under the control of the computer. At the same time, the galvanometer scans the photoelectric target point by point to generate training data. For the 2D moving platform, the *X*-axis moves from −500 mm to 500 mm every 10 mm in sequence, with a total of 101 steps. The Z-axis position moves from 320 mm to −320 mm every 10 mm, with a total of 65 steps. The photoelectric measurement system controls the galvanometer to track photoelectric target according to the error, $\delta (u_{p},v_{p})$, between the spot center and the photoelectric target center. The system synchronously records the measurement value $x_{i}={\left[D_{i},\theta_{xi},\theta_{zi}\right]}^{T}$ and spot center position $t_{i}={\left[x_{i}^{P},y_{i}^{P},z_{i}^{P}\right]}^{T}$ synchronously. A total of 6565 data are collected in the whole calibration plane and the training set $\left\{(x_{i},t_{i})|i=1,2,\ldots ,6565\right\}$ is obtained.

According to the KELM algorithm introduced in Section 3, the appropriate regularization coefficient and the initial value of kernel function are selected and then the calibration model is established. In order to verify the calibration accuracy of the KELM algorithm, 10-fold cross validation is adopted, the training data set is divided into ten subsets, and each subset is tested once. The other nine subsets are used as training sets and a total of ten training results are obtained.

For $x_{i}={\left[D_{i},\theta_{xi},\theta_{zi}\right]}^{T}$ in the testing data set $\left\{(x_{i}^{t},t_{i}^{t})|i=1,2,...65\right\}$, through the established calibration model, the calculation result $t_{i}^{Model}$ can be output and the root mean squared error (RMSE) in Equation (24) can be used to estimate the precision of calibration model as follows:(24)$$RMSE=\sqrt{\frac{1}{N}\sum_{i}^{N}{\left(t_{i}^{Model}-t_{i}\right)}^{2}}.$$

Taking the average RMSE of ten training models as the final RMSE, the regularization coefficient and kernel function parameters are adjusted continuously until the optimal RMSE value is obtained. Obviously, the smaller the RMSE value is, the higher the training accuracy will be.

### 4.2. Circle Testing Data Set and Verification

#### 4.2.1. Circle Testing Data Set

As shown in Figure 5, to verify the training effect and avoid over-fitting, the curve $C_{1}$ is designed, which is a circle in $O_{1}X_{1}Z_{1}$, with a center point $O_{1}$ and a radius of 320 mm. If $P_{a}$ is the point on the curve $C_{1}$, 950 points are uniformly selected on the circumference to constitute the circle testing data set. The position coordinate of $P_{a}$ can be represented as follows:(25)$$\{\begin{array}{c} x=320\cos\alpha \\ z=320\sin\alpha \end{array}(\alpha =i\cdot \frac{\pi }{250}-2\pi ,i=1,2,\ldots ,950).$$

The 2D moving platform is set at position I and it is controlled by the NC program to drive the photoelectric target moving along the 950 points in sequence. The measurement system can synchronously record each value, $x_{i}^{P_{a}}={\left[D_{i}^{P_{a}},\theta_{xi}^{P_{a}},\theta_{zi}^{P_{a}}\right]}^{T}$, to establish the circle test data set $\left\{(x_{i}^{\mathrm{cir}})|i=1,2,\ldots ,950\right\}$.

#### 4.2.2. Verification

For any measurement value, $x_{i}^{P_{a}}={\left[D_{i}^{P_{a}},\theta_{xi}^{P_{a}},\theta_{zi}^{P_{a}}\right]}^{T}$, in the test data set $\left\{(x_{i}^{\mathrm{cir}})|i=1,2,\ldots ,950\right\}$, on the basis of the calibration model of the galvanometer, the coordinate value $t_{i}^{Model\_P_{a}}={\left[x_{i}^{M\_P_{a}},y_{i}^{M\_P_{a}},z_{i}^{M\_P_{a}}\right]}^{T}$ of point $P_{i}$ is calculated according to the step in Part B of Section 3.

As shown in Figure 6, the difference between the actual radius at point $P_{i}$ and theoretical radius of $C_{1}$ is defined as the position error $\varepsilon_{i}^{\mathrm{cir}}$. The root mean square error (RMSE) in Equation (27), mean absolute error (MAE) in Equation (28), and the standard deviation (S_d_) in Equation (28) are used to measure the accuracy of the calibration model, as follows:(26)$$\varepsilon_{i}^{\mathrm{cir}}=\sqrt{{\left(x_{i}^{M\_P_{a}}\right)}^{2}+{\left(z_{i}^{M\_P_{a}}\right)}^{2}}-320\;(i=1,2,\ldots ,950),$$
(27)$$RMSE=\sqrt{\frac{1}{N}\sum_{i=1}^{N}\varepsilon_{i}^{\mathrm{cir}}{}^{2}},$$
(28)$$MAE=\frac{1}{N}\sum_{i=1}^{N}\left|\varepsilon_{i}^{\mathrm{cir}}\right|,$$
(29)$$S_{d}=\sqrt{\frac{1}{N}\sum_{i=1}^{N}{\left(\varepsilon_{i}^{\mathrm{cir}}-{\bar{\varepsilon }}_{i}^{cir}\right)}^{2}}$$
where ${\bar{\varepsilon }}_{i}^{\mathrm{cir}}$ is the average of $\varepsilon_{i}^{\mathrm{cir}}$.

The smaller the values of RMSE, MAE, and S_d_ are, the smaller the dispersion will be. This means that the data of the calibration result has lower noise and higher calibration accuracy.

#### 4.2.3. Radius Error

According to the calculated value $t_{i}^{Model\_P_{a}}={\left[x_{i}^{M\_P_{a}},y_{i}^{M\_P_{a}},z_{i}^{M\_P_{a}}\right]}^{T}$ of point $P_{i}$, the least squares method is used to fit the measured curve $C_{1}\prime$, ΔR is the difference between the radius $R_{C_{1}\prime }$ of curve $C_{1}\prime$ and the theoretical radius of $C_{1}$, and ΔR can be used as another indicator to measure the accuracy of the calibration model. If the center coordinate of $C_{1}\prime$ is (A,B), then the equation of $C_{1}\prime$ is:(30)$${\left(x_{i}^{M\_P_{a}}-A\right)}^{2}+{\left(z_{i}^{M\_P_{a}}-B\right)}^{2}=R_{C_{1}\prime }{}^{2}.$$

According to the least squares principle, the following equation can be obtained:(31)$$\{\begin{array}{l} \sum 2(x_{i}^{M\_P_{a}}{}^{2}+z_{i}^{M\_P_{a}}{}^{2}+ax_{i}^{M\_P_{a}}+bz_{i}^{M\_P_{a}}+c)x_{i}^{M\_P_{a}}=0\\ \sum 2(x_{i}^{M\_P_{a}}{}^{2}+z_{i}^{M\_P_{a}}{}^{2}+ax_{i}^{M\_P_{a}}+bz_{i}^{M\_P_{a}}+c)z_{i}^{M\_P_{a}}=0\\ \sum 2(x_{i}^{M\_P_{a}}{}^{2}+z_{i}^{M\_P_{a}}{}^{2}+ax_{i}^{M\_P_{a}}+bz_{i}^{M\_P_{a}}+c)=0 \end{array}.$$
By solving the equation, the intermediate parameters a, b, and c are obtained and the radius $R_{C_{1}\prime }$ can be represented as follows:(32)$$R_{C_{1}\prime }=\frac{I}{2}\sqrt{a^{2}+b^{2}-4c}.$$

The radius error can be represented as:(33)$$\Delta R=R_{C_{1}\prime }-320.$$

### 4.3. Sinc Testing Data Set and Verification

As shown in Figure 5, both the training data set and the circle testing data set are located in the calibration plane at position I. In order to verify the calibration effect of measurement data in other position, the “Sinc” is designed in the $O_{2}X_{2}Z_{2}$ plane. Taking into account the range of the 2D moving platform, we adjust the parameters of the Sinc and record it as $C_{2}$. The 2D moving platform is placed at position II, the plane of $O_{2}X_{2}Z_{2}$ is parallel to the coordinate plane, and the Sinc testing data set can then be established. The equation of curve $C_{2}$ represented as follows:(34)$$Z=\{\begin{array}{l} a\cdot \sin(b\cdot x)/x\;a=35000,b=1/70,x\neq 0\\ 500,\;x=0. \end{array}$$

If $P_{b}$ is the point on curve $C_{2}$ and 750 points are uniformly taken on curve $C_{2}$, then the coordinates of $P_{b}$ are represented as the following:(35)$$\{\begin{array}{l} x=140\alpha \\ z=\{\begin{array}{ll} 250\frac{\sin(2\alpha )}{\alpha }, & \alpha \neq 0\\ 500, & \alpha =0 \end{array} \end{array}\;(\alpha =i\cdot \frac{\pi }{250}-1.5\pi ,i=1,2,\ldots ,750).$$

The 2D moving platform is controlled by the NC program to drive the photoelectric target moving along 750 points in sequence. The system synchronously records each coordinate $x_{i}^{P_{b}}={\left[D_{i}^{P_{b}},\theta_{xi}^{P_{b}},\theta_{zi}^{P_{b}}\right]}^{T}$ to establish the Sinc testing data set $\left\{(x_{i}^{\sin c})|i=1,2,\ldots ,750\right\}$.

In Figure 5, the projection curve of $C_{2}$ in the plane $O_{1}X_{1}Z_{1}$ is $C_{3}$ and the projection point corresponding to point $P_{b}$ is $P_{c}$ For any measurement value $x_{i}^{P_{b}}={\left[D_{i}^{P_{b}},\theta_{xi}^{P_{b}},\theta_{zi}^{P_{b}}\right]}^{T}$, in terms of the calibration model of the galvanometer, the coordinate value $t_{i}^{Model\_P_{c}}={\left[x_{i}^{M\_P_{c}},y_{i}^{M\_P_{c}},z_{i}^{M\_P_{c}}\right]}^{T}$ of point $P_{c}$ is calculated according to the steps in Part B of Section 3.

In accordance with the geometric relationship in Figure 5, given the coordinates $P_{b}(x_{P_{b}},y_{P_{b}},z_{P_{b}})$ and $P_{c}(x_{P_{c}},y_{P_{c}},z_{P_{c}})$, the following relationship exists:(36)$$\frac{OO_{1}}{OO_{2}}=\frac{x_{P_{c}}}{x_{P_{b}}}=\frac{z_{P_{c}}}{z_{P_{b}}}=\frac{1.2}{2.4}.$$

Therefore, the calibration value of point $P_{b}$ can be calculated by Equation (37) as follows:(37)$$t_{i}^{Model\_P_{b}}=\left[\begin{array}{c} x_{i}^{M\_P_{b}}\\ y_{i}^{M\_P_{b}}\\ z_{i}^{M\_P_{b}} \end{array}\right]\approx \left[\begin{array}{l} 2.0417\cdot x_{i}^{M\_P_{c}}\\ 2.45\\ 2.0417\cdot z_{i}^{M\_P_{c}} \end{array}\right](i=1,2,\ldots ,750).$$

As shown in Figure 7, the difference between the calibrated and theoretical value of the $P_{b}$ is defined as position error $\varepsilon_{i}^{\sin c}$, which is used to measure the accuracy of the calibration model of the galvanometer. The expressions for $\varepsilon_{i}^{\sin c}$ can be represented as follows:(38)$$\varepsilon_{i}^{\sin c}=z_{i}^{M\_P_{b}}-35000\sin(x_{i}^{M\_P_{b}}/70)/x_{i}^{M\_P_{b}}(i=1,2,\ldots ,750).$$

As with the circle testing data set, the calibration effect of the Sinc testing data set is measured by accuracy indicators such as RMSE, MAE, and S_d_ in Equations (27)–(29).

## 5. Experimental Verification

The performance of four KELM methods is analyzed and compared with the physical model method and original ELM in this section. All these algorithms are run on R2014a MATLAB software, which is installed in a personal computer with 3.2 GHz CPU and 8.0 GB RAM. The training set has 6565 points, the circle testing data set has 950 points, and the Sinc testing data set has 750 points.

After the optimization training, the optimized parameters of several data-driven methods are shown in Table 1.

### 5.1. Validation Results

#### 5.1.1. Results of the Circle Testing Data Set

The four kernel functions are shown in Equations (20)–(23). They are used as the kernel functions of KELM algorithm, respectively, and the calibration accuracy is verified by using the circle testing data set. The optimal training results and the training time of each algorithm are compared with the physical model method and original ELM. The position error $\varepsilon_{i}^{\mathrm{cir}}$ is calculated according to Equation (26) and the RMSE value of each method is calculated according to Equation (27). The mean absolute error (MAE) and standard deviation (S_d_) of each method are calculated by Equations (28)–(29), and the fitting radius and radius error ΔR are calculated by Equations (32)–(33). The results of the various methods are shown in Table 2.

It can be seen from Table 1 that the Mexican Hat wavelet KELM is observed to attain the most optimal results among all the models, with a radius of 320.03 mm and ΔR of 0.03 mm, RMSE value of 0.4130 and MAE of 0.3259, and supported by small standard deviation at 0.2536. This indicates that the accuracy of wavelet KELM is higher than that of other methods. Figure 8 shows the four calibrated curves of the circle. In terms of the $\varepsilon_{i}^{\mathrm{cir}}$ values of each method, the position error is shown in Figure 9. It can be seen from Figure 8 and Figure 9 that the calibrated curves are close to the theoretical circle. The calibration accuracy of the two wavelet KELM is higher than the original ELM and physical model method. In terms of the two wavelet kernel functions, the Mexican Hat wavelet KELM is slightly better than the Morlet wavelet KELM. Compared with original ELM, the calibration effect of the Mexican Hat wavelet KELM is improved. The RMSE, MAE, and S_d_ are reduced by 16.4%, 14.6%, and 19.2%, respectively.

#### 5.1.2. Results of Sinc Testing Data Set

The calibrated model is verified by using the Sinc testing set and the results of four different kernel functions are obtained. As shown in Table 3, the results are compared with the physical model and the original ELM, respectively.

According to Equation (37), the positions of curves $C_{2}$ are calculated and the calibrated curves, corresponding to the four KELM, are drawn in Figure 10. The position error $\varepsilon_{i}^{\sin c}$ is calculated according to Equation (38), the RMSE value of each method is calculated according to Equation (27), and the mean absolute error (MAE) and standard deviation (S_d_) of each method are calculated by Equations (28)–(29). The calculation results are listed in Table 3 and the position error of the four KELM are shown in Figure 11.

As can be seen from Table 3 and Figure 10 and Figure 11, the accuracy of the wavelet KELM is higher than that of other methods. It is observed to attain the best RMSE among all other models, with an RMSE value of 1.1695, MAE of 0.9777, and S_d_ of 0.5495. This indicates that the accuracy of the Mexican Hat wavelet KELM is slightly better than the Morlet wavelet KELM and has better results than the other methods. Compared with the original ELM, the RMSE, MAE, and Sd are reduced by 38.6%, 39.4%, and 36.6%, respectively.

#### 5.1.3. Comparison of Calculation Speed

In terms of the calculation speed of the algorithm, the offline model training time, testing time, and online correction time are compared, respectively. Each index is tested 20 times and the average time is taken as the final value, which is listed in Table 4. According to the procedure flow in Section 3.3, Part B: Step (1) to Step (4), the online calculation time of calibration algorithm is calculated and the results are listed in the last column of Table 4.

As listed in Table 4, because of the simplicity of the algorithm and the small amount of computation, the calculation time of the physical model method is the fastest, the original ELM method is the second, and the kernel-based calibration methods are the slowest. This is due to the complexity of the kernel matrix $\Omega_{Test}$ and output weight $O_{w}$, which causes the calculation time to be longer and slows down the calculation speed. As a result, the offline training and testing time of KELM are both greater than original ELM. The offline training time of the Morlet wavelet KELM is the longest, which reaches 4.76 s.

In the process of online calculation of the kernel-based method, the offline model training has been completed and the output weight $O_{w}$ has been obtained. For this reason, the computation of the online calibration algorithm is reduced. In addition, the online measurement data is one-dimensional data, which greatly reduces the amount of calculation. This is another reason why the operation time of the online calibration algorithm is greatly shortened. Among the two wavelet KELM with high calibration accuracies, the online calibration time of the Morlet wavelet KELM is less than 0.42 ms and the Mexican Hat wavelet KELM is less than 0.39 ms.

### 5.2. Analysis

According to the experimental process, it can be seen that the calibration data of the galvanometer contains a variety of nonlinear factors and noises, such as installation errors, system control noise, laser ranging noise, angle measurement noise, and other random noises, etc., which is typical of a nonlinear sample set. The magnitude and characteristics of these errors and noises are unknown.

The results from Section 5.1 indicate that the accuracy of KELM algorithm with different kernel functions differs greatly. Among the four kernel functions, the error of the Gaussian kernel function is the largest, even worse than original ELM. The two wavelet KELM are higher than the original ELM and other kernel functions. The Mexican Hat wavelet kernel is slightly higher than the Morlet wavelet kernel. The smaller RMSE value means that the wavelet KELM has better generalization ability than other calibration methods for the nonlinear samples of the galvanometer.

Further analysis of the reasons for this result shows that the wavelet kernel is constructed from a wavelet function, which can form a set of primary functions of space $L^{2}(R)$ through stretching and translation. For any nonlinear function, it can be expressed as a linear combination of the primary functions and has a good fitting ability for nonlinear functions. Therefore, wavelet KELM has a better generalization ability for nonlinear samples than other kernel ELM methods and the original ELM.

As can be seen from Table 1 and Table 2, for the two testing data sets, the error of the physical model method has a large MAE value and S_d_ value, which means that the calibration result contains a large noise, while the error dispersion of ELM and wavelet KELM is greatly reduced. This means that both ELM and wavelet KELM method have a certain noise suppression ability, but the wavelet KELM has better generalization ability and noise suppression effect than other methods.

From Equation (23), it can be found that the Mexican Hat wavelet kernel $\kappa (x_{i},x_{j})$ is a function of the distance between vector $x_{i}$ and vector $x_{j}$. With the increase of the distance between the two vectors, the influence of $x_{j}$ on $x_{i}$ decreases. The attenuation law is determined by the Mexican Hat wavelet kernel function and is significantly affected by σ. The larger the value σ is, the faster the weaken speed of $\kappa (x_{i},x_{j})$ is. It can also be seen from Equation (17) that the output of the calibration model is obtained by the superposition of the product of the Mexican Hat wavelet kernel $\kappa (x_{i},x_{j})$ and the weight $O_{w}$ at various positions. In this process, the Mexican Hat wavelet kernel is like a low-pass filter, which weakens the influence of noise points, so that the wavelet KELM has low MAE and S_d_ values and a noise suppression effect.

After the wavelet KELM is adopted, the complexity of the algorithm increases and the speed of the calibration algorithm slows down. For the online calibration algorithm, the operation time of the Mexican Hat wavelet KELM is slightly faster than that of the Morlet wavelet KELM. As can be seen from the comparison between Equations (22) and (23), the independent variable of the Mexican Hat wavelet kernel only contains quadratic terms and the Morlet wavelet kernel also contains the primary term and its cosine operation. For the higher-dimensional input vector $x_{i}$ and $x_{j}$, the calculation of $\left\|x_{i}-x_{j}\right\|$ requires decomposing it into the product form of multiple one-dimensional vectors, then performing the cosine calculation, which will take a certain calculation time and makes Morlet wavelet kernel functions slower than the Mexican Hat wavelet kernel functions.

Based on the above analysis, the Mexican Hat wavelet KELM has higher accuracy than other existing methods for 2D galvanometer calibration. Although the calculation speed is slower than the original ELM, it can still reach 0.39 ms, which can meet the needs of a conventional online measurement system.

## 6. Conclusions

In this paper, we have proposed an online calibration method that can balance calibration accuracy and calculation speed. Wavelet KELM algorithm is adopted to establish the mapping model $f:x_{i}\to t_{i}$ between the measurement data and the spot position data. Based on this, an online calibration model is built. Through experiments, the training data set and two groups of testing data sets are obtained. The calibration effects of the physical model method, original ELM, Polynomial KELM, Gaussian KELM, Morlet wavelet KELM, and Mexican Hat wavelet KELM are compared in terms of both accuracy and speed. The results show that the accuracy of the data-driven calibration methods is higher than that of the physical model. For the KELM algorithms, the calibration effects of different kernel functions vary greatly. The calibration accuracy of the Morlet wavelet KELM and Mexican Hat wavelet KELM is obviously higher than the other four methods, with the highest calibration accuracy for the latter. In terms of algorithm speed, although the online calibration speed of the wavelet KELM is slower than that of original ELM, it can still meet certain real-time calculation requirements. The real-time computation times of the Mexican Hat wavelet KELM and Morlet wavelet KELM are less than 0.39 ms and 0.49 ms, respectively. The results show that the Mexican Hat wavelet KELM is more suitable for the online calibration of a galvanometer.

## Figures and Tables

**Figure 1 sensors-19-01353-f001:**
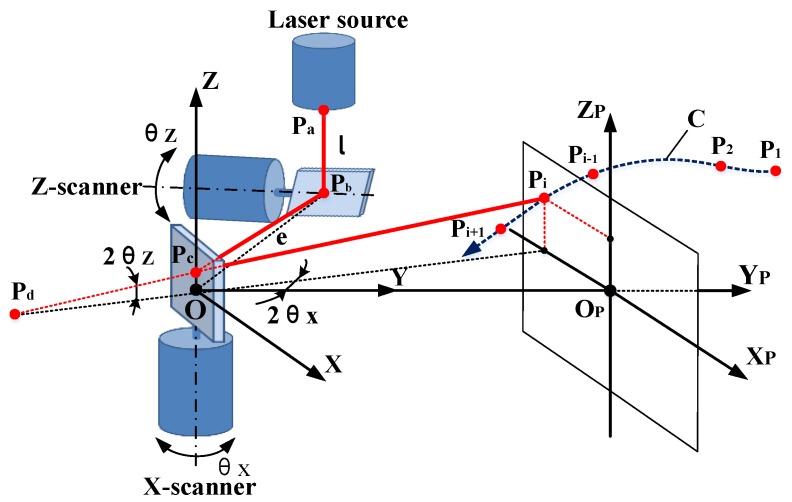
Principle of a photoelectric measurement system.

**Figure 2 sensors-19-01353-f002:**
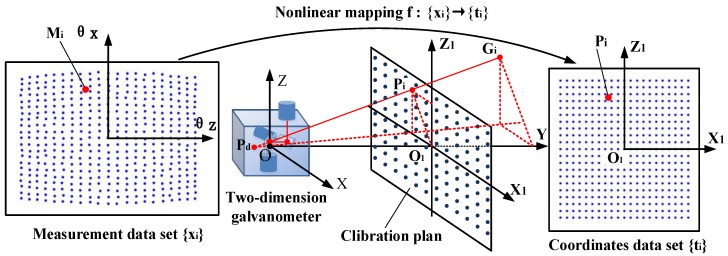
Principle of data-driven calibration.

**Figure 3 sensors-19-01353-f003:**
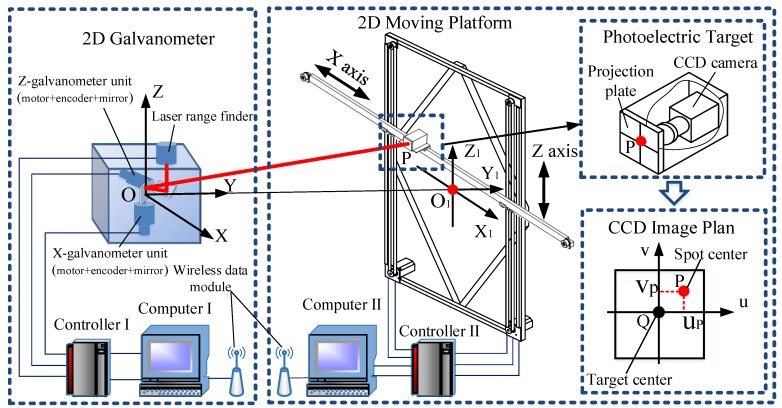
Experimental system configuration.

**Figure 4 sensors-19-01353-f004:**
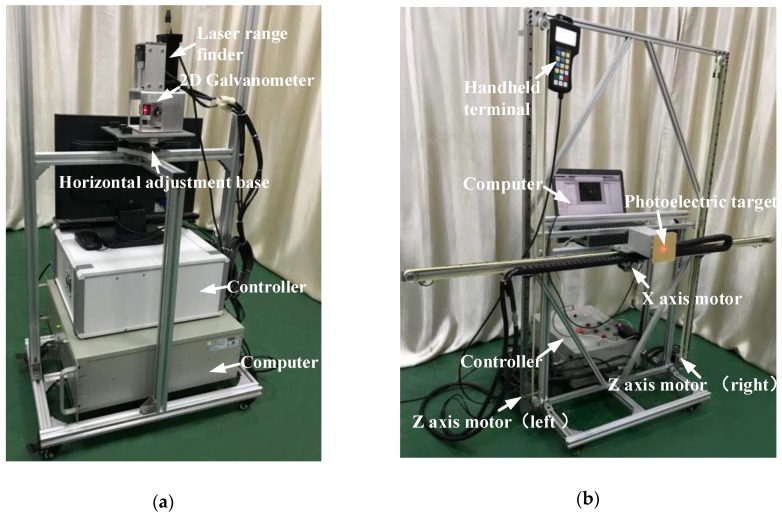
Experimental setup of galvanometer calibration (**a**) 2D galvanometer system; (**b**) 2D moving platform and photoelectric target.

**Figure 5 sensors-19-01353-f005:**
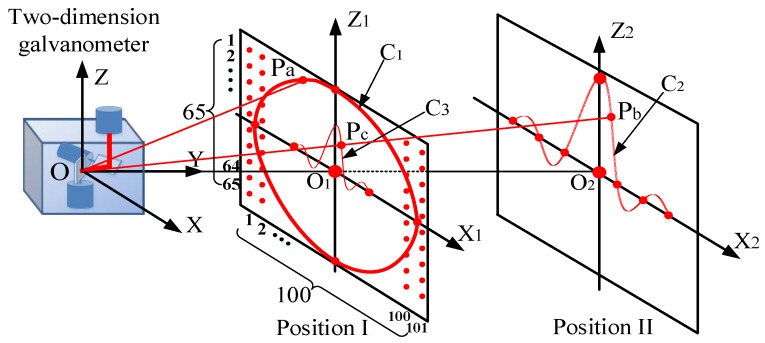
Calibration data generation.

**Figure 6 sensors-19-01353-f006:**
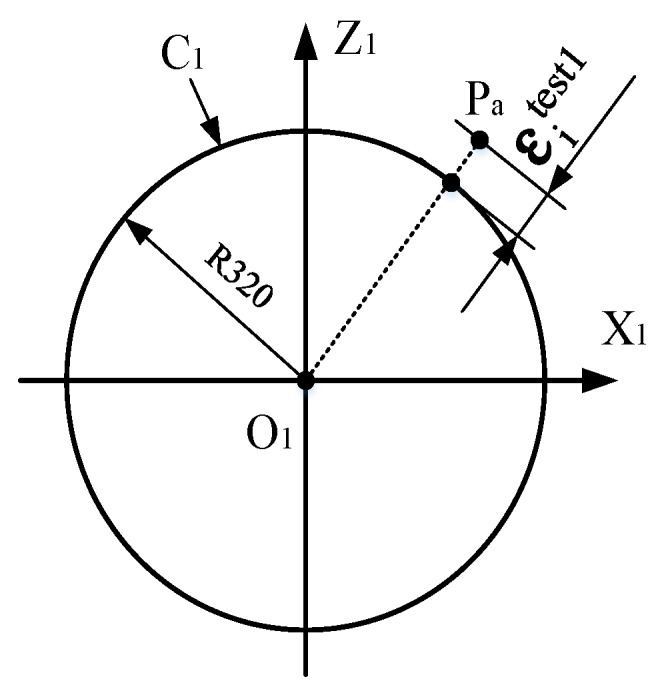
Position error between the actual and theoretical radius.

**Figure 7 sensors-19-01353-f007:**
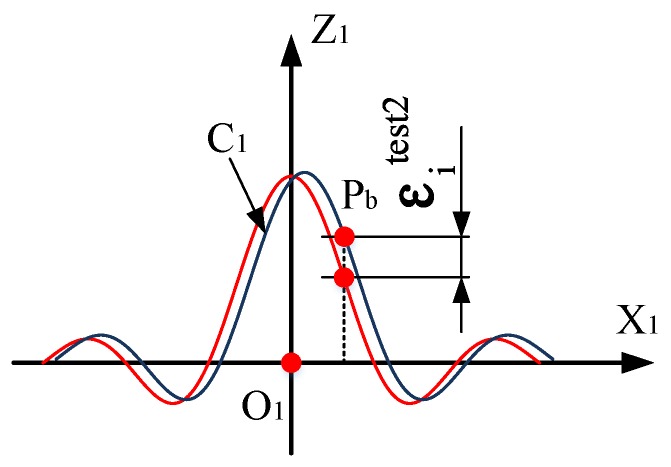
Position error between the calibrated and theoretical value.

**Figure 8 sensors-19-01353-f008:**
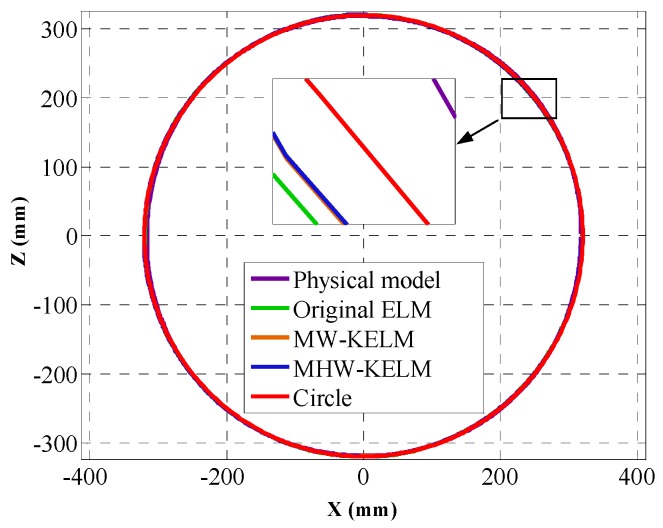
Calibrated circle.

**Figure 9 sensors-19-01353-f009:**
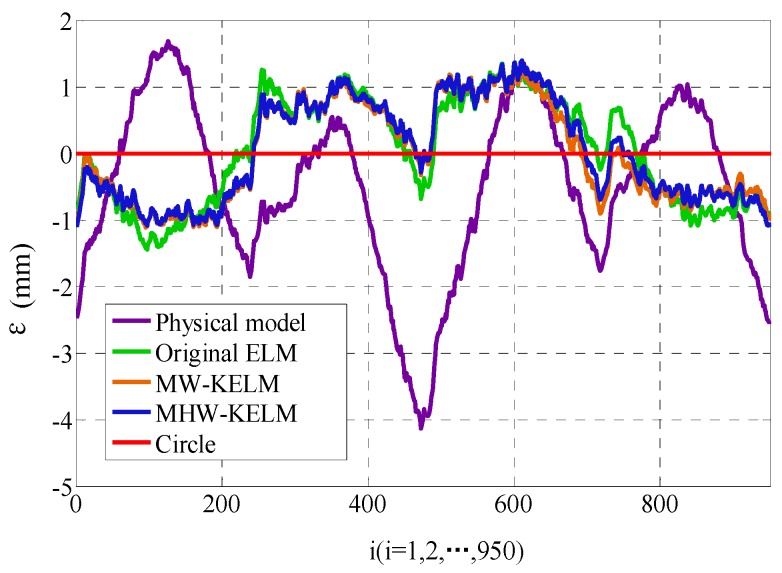
Position error of the calibrated circle.

**Figure 10 sensors-19-01353-f010:**
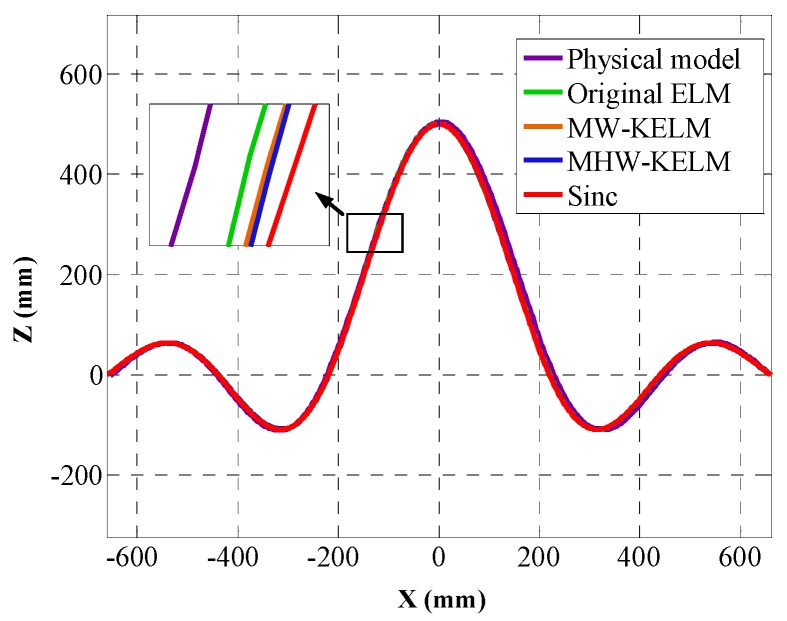
Calibrated Sinc.

**Figure 11 sensors-19-01353-f011:**
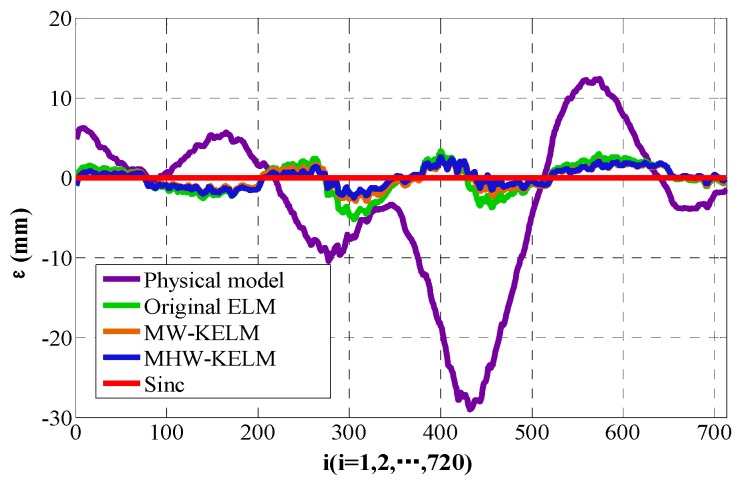
Position error of calibrated Sinc.

**Table 1 sensors-19-01353-t001:** Optimized parameters of the data-driven method.

Method	Optimized Parameter	Training Data Set
Original ELM	L=600	$\left\{(x_{i},t_{i})|i=1,2,\ldots ,6565\right\}$ with 6565 samples.
Polynomial-KELM	C = $2^{40}$, c=0.001, d=4	
Gaussian-KELM	C = $2^{30}$, σ=0.022	
Morlet wavelet (MW)-KELM	C = $2^{38}$, a=355, σ=14.14	
Mexican Hat wavelet (MHW)-KELM	C = $2^{37}$, σ=9.80	

**Table 2 sensors-19-01353-t002:** Results of the circle testing data set.

Method	R (mm)	ΔR (mm)	RMSE	MAE	S_d_
Physical model	319.45	−0.55	1.7109	1.2954	1.1176
Original ELM	320.04	0.04	0.4941	0.3817	0.3138
Polynomial-KELM	320.04	0.04	0.4794	0.3777	0.2953
Gaussian-KELM	320.05	0.05	0.6189	0.4956	0.3707
Morlet wavelet (MW)-KELM	320.03	0.03	0.4371	0.3479	0.2646
Mexican Hat wavelet (MHW)-KELM	**320.03**	**0.03**	**0.4130**	**0.3259**	**0.2536**

**Table 3 sensors-19-01353-t003:** Results of Sinc testing data set.

Method	RMSE	MAE	S_d_
Physical model	10.0716	7.3047	5.9375
Original ELM	1.9045	1.6135	0.8664
Polynomial-KELM	1.7992	1.5459	0.7882
Gaussian-KELM	2.4589	1.8765	1.3608
Morlet wavelet (MW)-KELM	1.2581	1.0759	0.5583
Mexican Hat wavelet (MHW)-KELM	**1.1695**	**0.9777**	**0.5495**

**Table 4 sensors-19-01353-t004:** Calculating time.

Method	Training	Circle	Sinc	Real-Time (ms)
	Training Time (s)	Testing Time (s)	Testing Time (s)	
Original ELM	0.8103	0.0143	0.0123	<0.03
Polynomial-KELM	3.8273	0.2944	0.2214	<0.49
Gaussian-KELM	3.3123	0.1941	0.1606	<0.35
Morlet wavelet (MW)-KELM	4.3145	0.3651	0.2641	<0.42
Mexican Hat wavelet (MHW)-KELM	3.9992	0.2095	0.1816	<0.39

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
