# Peer review of "An Online Calibration Method for a Galvanometric System Based on Wavelet Kernel ELM"

_sensors, 2019, doi:10.3390/s19061353_

Reviewer 1 Report

The manuscript present an interesting idea on galvanometer calibration or modeling problem. However, following issues must be settled before its publication. 

The quality of English is such that the manuscript cannot to be accepted in its present form.

Some key concept is not clearly explained in the manuscript e.g. "Physical model method" and what is the difference between it with ELM. Is it means an essential galvanometer model difference? or it only means a training algorithm difference, which only affect the parameter in galvanometer modeling but not the modeling itself?. e.g assume the control voltage of gal is generated as a+bx. In ELM and physical model result, the only difference is a an b value rather than the former use nonlinear function f(x) instead. This is not clearly declared in the manuscript but is very critical to evaluate the impact of this research. 

The author claim their method as a "online" calibration method and "calculation speed" is one of the major advantage. The major bottleneck for this calibration method is the training data generation time. Compared with at least minutes (estimated from the hardware pic in the manuscript), ms reduction on calculation time is no meaning at all. The accuracy improvement (if proved) is good enough and no need overclaim it.

The details of the training process of the different algorithm are not provided. e.g. how the training is executed, how many iteration or samples used in the training process for different algorithms, are they same and how to evaluate the training result or any correction are used in the process etc. These details are critical for the judgement of the fairness in following comparison.  

In summary, I am not support the acceptance of paper based on its current form. At least a major revise is need to make it clear and understandable, especially on the critical issues motioned above.

Author Response

First, I would like to thank the reviewers for their affirmation of the content of this paper, and at the same time, I would also like to thank the reviewers for their valuable comments on this paper. The study of online calibration method may not be clearly stated in this paper. We will explain it further in the revised version. The revised draft is in the attachment for your review. Thank you!

Reviewer 2 Report

The paper presents the use of wavelet kernel functions for galvanometer calibration using a neural network (the so-called extreme learning network).

The paper fits well within the scope of the journal Sensors. It is well-written with clear graphical illustrations. The text needs some minor English language revisions, but overall the writing style is good.

Concerning the scientific contributions reported in the paper I have some major concern which I think should be resolved before the paper can be accepted for publication:

-        The authors claim that the method allows an on-line real-time calibration. Firstly, I have the impression that this is also the case for the other calibration algorithms mentioned in the paper. Secondly, I do not see why a real-time calibration is needed. For a certain galvanometer system the calibration should be done only once and hence the calibration time is not critical.

-        The fact that wavelets are noise insensitive is stressed in the paper. However, there is no information about measurement noise in the paper. I would at least expect some numbers on the noise variance of the input and output of the performed validation experiments.

-        I am not really impressed by the results of the proposed method compared to the ELN technique. This does not seem a significant contribution to me.

-        It is not mentioned how the wavelet basis functions are selected (e.g. at which scales?).

Author Response

First, I would like to thank the reviewers for their affirmation of the content of this paper, and at the same time, I would also like to thank the reviewers for their valuable comments on this paper. The study of online calibration method may not be clearly stated in this paper. We will explain it further in the revised version. The revised draft is in the attachment for your review. Thank you!

Round  2

Reviewer 1 Report

The revised manuscript has significant improvement compared with its original version. However, following issues need to be settled before the publication.

I think that "online" calibration is over-claimed. According to paper, only single calibration plane O1X1Z1 in Fig.2 will need to be setup and obtained G_i coordinate will be used for 3D (as it is claimed as coordinator) measurement on targeted sample. Therefore, only one time calibration is needed. Unless, the G_i coordinator only used to measure the surfaces of the objects normal to the OZ. For other surfaces with different orientation , a new calibration plane O2X2Z2 need to be generated. However, in the later case, the training data generation may need to done for all possible orientation of the optical axes. Only in this case, decreasing the algorithm training time will reduce the total calibration time (1msX65651). But claim it as "online" method may be quite misleading. At least, the definition and need of so call "online" need to be clearly explained.             

 It seems the grid generated by moving object is used as the ideal calibration standard (calibration plane O1X1Z1) in the paper. The motors used to moving the object is assumed to be perfect without position error in the discussion. If it is true, this assumption should be highlighted in the paper. Also the feedback loop is used to control the galvanometer rotation motor not moving object motor, which is quite confused without declaration of above assumption.

The English writing and the utlization of symbol need to be revised. e.g. line 69-71  "Formula (5) shows that, given the measured value (G zi G xi G Mi Di) , according to the value Pi can be found in the calibration plane O x z , the coordinates of  Gi can be obtained by the coordinates P (x , y ,z ) t and the distance value p Di ."   It is really hard to understanding the logic of "given Mi, according to Mi, Pi can be found, Gi can be obtained by Pi" within a single sentence full of comma and mapping them to the experiments described later. Some symbols, e.g."l" in (1,4,5),are used without definition.          

Author Response

First of all, we would like to thank the reviewers for their valuable comments, and put forward many valuable suggestions for the revision of the article. We have entrusted the English editor of MDPI to modify the English language and style of the manuscript, and further improved the suggestions put forward by reviewers. Please review the revised manuscript again. Thank you very much! Thank you!

Reviewer 2 Report

The authors have satisfactorily addressed all my comments.

Author Response

(The authors gave the same response as above.)
